# Glioma Stem Cells in Pediatric High-Grade Gliomas: From Current Knowledge to Future Perspectives

**DOI:** 10.3390/cancers14092296

**Published:** 2022-05-04

**Authors:** Marc-Antoine Da-Veiga, Bernard Rogister, Arnaud Lombard, Virginie Neirinckx, Caroline Piette

**Affiliations:** 1Laboratory of Nervous System Disorders and Therapy, GIGA Institute, University of Liège, 4000 Liège, Belgium; madaveiga@uliege.be (M.-A.D.-V.); bernard.rogister@uliege.be (B.R.); alombard@chuliege.be (A.L.); virginie.neirinckx@uliege.be (V.N.); 2Department of Neurology, CHU of Liège, 4000 Liège, Belgium; 3Department of Neurosurgery, CHU of Liège, 4000 Liège, Belgium; 4Department of Pediatrics, Division of Hematology-Oncology, CHU Liège, 4000 Liège, Belgium

**Keywords:** pediatric high-grade glioma, diffuse midline glioma, glioblastoma, diffuse intrinsic pontine glioma, cancer stem cell, glioma stem cell, glioma initiating cell, subventricular zone

## Abstract

**Simple Summary:**

Pediatric high-grade glioma (pHGG) has a dismal prognosis in which the younger the patient, the more restricted the treatments are, in regard to the incurred risks. Current therapies destroy many tumor cells but fail to target the highly malignant glioma stem cells (GSCs) that adapt quickly to give rise to recurring, treatment-resistant cancers. Despite a lack of consensus around an efficient detection, GSCs are well described in adult brain tumors but remain poorly investigated in pediatric cases, mostly due to their rarity. An improved knowledge about GSC roles in pediatric tumors would provide a key leverage towards the elimination of this sub-population, based on targeted treatments. The aim of this review is to sum up the state of art about GSCs in pHGG.

**Abstract:**

In children, high-grade gliomas (HGG) and diffuse midline gliomas (DMG) account for a high proportion of death due to cancer. Glioma stem cells (GSCs) are tumor cells in a specific state defined by a tumor-initiating capacity following serial transplantation, self-renewal, and an ability to recapitulate tumor heterogeneity. Their presence was demonstrated several decades ago in adult glioblastoma (GBM), and more recently in pediatric HGG and DMG. In adults, we and others have previously suggested that GSCs nest into the subventricular zone (SVZ), a neurogenic niche, where, among others, they find shelter from therapy. Both bench and bedside evidence strongly indicate a role for the GSCs and the SVZ in GBM progression, fostering the development of innovative targeting treatments. Such new therapeutic approaches are of particular interest in infants, in whom standard therapies are often limited due to the risk of late effects. The aim of this review is to describe current knowledge about GSCs in pediatric HGG and DMG, i.e., their characterization, the models that apply to their development and maintenance, the specific signaling pathways that may underlie their activity, and their specific interactions with neurogenic niches. Finally, we will discuss the clinical relevance of these observations and the therapeutic advantages of targeting the SVZ and/or the GSCs in infants.

## 1. Pediatric High-Grade Gliomas: From Histologic to Histomolecular Classification

Brain and other central nervous system (CNS) tumors represent the most frequent solid malignant neoplasms in people aged 0–19 years [1]. Among them, gliomas form the most frequent subgroup and account for 51.6% and 31.1% of CNS tumors in children (0–14 years) and adolescents (15–19 years), respectively [1]. As a group, high-grade gliomas (HGG) constitute the most common pediatric malignant tumor of the CNS, with an incidence of 0.87 per 100,000 children in the United States [2].

For years, pediatric gliomas were grouped with their adult counterparts, despite known differences in various aspects. More recently, however, the identification of specific genetic abnormalities underlying pediatric gliomas has progressively allowed the recognition of prognostically and biologically distinct entities primarily occurring in children (but sometimes also in adults). The recent fifth edition of the WHO Classification of Tumors of the CNS (WHO CNS5) divided the *“Gliomas, Glioneuronal Tumors, and Neuronal Tumors”* entity into six families, including two pediatric types: (1) the *“Pediatric-type diffuse low-grade gliomas”* (pLGG), expected to have good prognosis and (2) the *“Pediatric-type diffuse high-grade gliomas”* (pHGG), predicted as more aggressive [3]. Noteworthy, the term glioblastoma (GBM) is no longer used in children and adolescents.

Among the pHGG, the WHO CNS5 classification distinguishes four subtypes, based on histopathological and molecular characteristics [4] (Figure 1):

(a) the *“Diffuse midline glioma (DMG), H3 K27-altered”* subgroup was first defined in the 2016 CNS WHO classification [4], based on the existence of a diffuse growth pattern, a midline location (e.g., thalamus, brain stem, or spinal cord) and a K27M mutation (lysine 27 is replaced by methionine) in the histone H3 (H3) genes *H3F3A* or *HIST1H3B* [5], leading to epigenetic changes [5,6,7,8,9,10]. In 2021, other alterations (e.g., EZHIP protein overexpression) were recognized as defining the same entity [11]. *DMG, H3 K27-altered* includes tumors previously called diffuse intrinsic pontine glioma (DIPG).

(b) the *“Diffuse hemispheric glioma, H3 G34-mutant”* entity is depicted by the presence of an arginine or valine residue (less frequent) at codon 34 instead of a glycine residue in the *H3F3A* and represents approximately 18% of the cortical pHGG in children [12]. Patients in this subgroup are mainly adolescents and young adults and have better overall survival (OS), compared to patients with K27 mutation [5].

(c) the *“Diffuse pediatric-type high-grade glioma, H3-wildtype and IDH-wildtype”* requires a molecular characterization and the integration of histopathological and molecular data for diagnostic purposes [13].

(d) the *“Infant-type hemispheric glioma”* occurs in newborns and infants and has a distinct molecular profile, with gene fusions involving *ALK*, *ROS1*, *NTRK1*/*2/3*, or *MET* [14].

Most of the epidemiologic data, as well as basic science and clinical studies, were reported before this recent WHO CNS5 classification and classically divided the pHGG between DIPG and non-DIPG tumors.

DIPG represents the major cause of death in children with CNS tumors. Male and female patients are equally affected and the median age at diagnosis is 7 years [15]. Outside clinical trials, the current standard of care is limited to focal radiotherapy (RT) [16]. Despite innovative therapeutic strategies [17], the prognosis remains poor with a median OS of 11.2 months [18].

Non-DIPG pHGG are equally distributed between male and female patients. The mean age at diagnosis is 8.7 years [19]. The current standard of care consists of surgical resection followed by RT and concomitant adjuvant chemotherapy. Despite a significant number of clinical trials for children with HGG [20], their prognosis did not improve over the last decades, with a median OS of 17 months [19].

For the purpose of the present review, we will adopt the recent histomolecular WHO CNS5 classification wherever possible. In case molecular data is not available, we will distinguish DMG from non-DMG pHGG wherever feasible, based on the location of the tumor. In case the location is not specified, we will use the broader category “pHGG” (Figure 1).

In conclusion, pHGG are now individualized from their adult counterpart, to reflect their specificities in terms of pathogenesis and prognosis. As a group, they represent the most common pediatric cancer of the CNS. Despite numerous innovative treatments, they remain the major cause of death in children with CNS tumors, requiring new lines of research [21,22].

## 2. From Stem Cells to Glioma Stem Cells

Adult stem cells are defined as unspecialized cells that own the property to self-renew and to differentiate into diverse specialized cell types [21]. Stem cells ensure the production of tissue-specific short-lived cells in the intestinal epithelium, the skin or the hematological system, while allowing these tissues to constantly self-renew [22,23]. In the adult CNS, neural stem cells (NSCs) are found in neurogenic niches where they were shown to differentiate into committed neural subtypes, such as neurons, astrocytes or oligodendrocytes [24].

Cells with stem-like properties are also found within cancerous tissues, where they are called cancer stem cells (CSCs). Generally, less than 5% of the malignant cells of a tumor mass are known to preserve a self-renewal potential through multiple generations and can proliferate and reproduce a complex new tumor [25]. The basis of the CSC theory appeared for the first time in 1937, when the injection of a single leukemia cell into mice produced a quickly lethal leukemia [26]. Over 50 years later, Dick and coworkers identified a subset of patient-derived leukemia cells that could traffic to the bone marrow of immunodeficient mice, to actively proliferate and maintain the original leukemic cell phenotype [27]. They further characterized leukemia-initiating cells and showed that they could differentiate in vivo and self-renew [28].

While there has never been any clear consensus regarding their definition, CSCs were classically associated with four major in vitro properties [29]: (1) formation of spherical colonies in suspension cultures [30,31,32,33], (2) expression of different levels and patterns of surface and intracellular markers [34], (3) ability to differentiate towards multiple lineages [30,35] and (4) higher resistance to radiation and/or chemotherapy than the non-CSCs tumoral cells [36,37,38,39]. Importantly, in vivo, CSCs have been described as able to initiate the development of new tumors in serial xenotransplantation experiments [30]. So far, CSCs have thus been defined by the combination of multiple operational criteria.

Since their initial description, CSCs were identified in several cancers, including breast cancer [25], prostate cancer [40], colorectal cancer [37,41,42] and pancreatic cancer [43].

In GBM, CSCs were first evidenced in 2002 in adults by Ignatova *et al*., who identified neural stem-like cells expressing astroglial and neuronal markers [44]. They were then demonstrated in pediatric pHGG by Hemmati et al., who identified some tumor cells expressing NSC and other stem cell-associated proteins, including CD133 [45] (see below). In 2004, Singh and collaborators reported the ability of adult GBM and pHGG CD133-positive cells to generate tumors that recapitulated the patient’s tumor histology in serial xenotransplantation [30]. Finally, in 2006, Bao *et al*. showed that CD133-positive glioma cells confer radioresistance and could be responsible for tumor recurrence after RT. They were the first to use the term “glioma stem cells” (GSCs) [46]. Since then, it has been estimated that approximately one GSC for 1000 tumor cells is present in adult GBM [47]. Gimple et al. proposed a GSC definition based on multiple functional criteria that assess the cell capacity to self-renew, initiate tumor through serial transplantation, and recapitulate tumor cell heterogeneity [48].

## 3. From Adult to Pediatric Glioma Stem Cells

Despite their initial description in pHGG as early as in 2004 [30], GSCs have been poorly characterized and their role has been under-researched in children as compared to adults. This can partly be explained by the low incidence of pHGG and the rarity of the pHGG tissue available for investigations. Here, we will review and summarize the literature that supports GSC identification in pHGG based on the putative stem cell protein markers (3.1) and illustrates the capacity of GSCs to recapitulate the initial patient tumor at the molecular or phenotypic levels (3.2). We will also discuss the more recent insight into GSC generation and maintenance in pHGG (3.3) (Figure 2). Although beyond the scope of the present review, note that CSCs have also been described in other childhood brain cancers, especially in embryonal tumors [30,35,49,50,51] and ependymomas [52,53,54,55]. In these tumors, many of the following molecular markers and phenotypic features were also used for CSC identification and characterization.

### 3.1. Putative Markers for GSCs in pHGG

As described above, adult GSCs have been defined based on their capacity to self-renew, differentiate, and form tumors. Several proteins and transcription factors were proposed as putative markers, although it is nowadays well-accepted that their expression lacks specificity and consistency, and does not allow the accurate isolation of GSCs from other tumor cells [57]. Nonetheless, the many studies that harnessed such markers to support the existence of GSCs in pediatric HGG definitely contributed to the current knowledge on pediatric GSCs, and these findings are described in the following section (Figure 2A).

#### 3.1.1. CD133

Prominin-1, also called CD133, is a pentaspan transmembrane glycoprotein, which functional roles in both normal and pathological conditions are yet unclear: it could be involved in membrane organization, act as a scaffolding protein, maintain stem cell-like properties and determine cellular fate (reviewed in Glumac et al. [58]). CD133 has been used to detect and isolate NSCs, and was suggested as an important contributor to their self-renewal and differentiation potential [59], although further research has demonstrated that CD133-negative NSCs were highly clonogenic and multipotent [60].

CD133 has been the most common cell surface antigen used to detect and isolate supposed CSCs in various types of solid tumors, including in gliomas (reviewed in Glumac et al. [58]). In pHGG, CD133 was first proposed in 2003 as an important protein for GSC discrimination [35,45]. Then, Singh et al. showed that only 100 CD133-positive pHGG cells were necessary to initiate a tumor upon intracranial transplantation into adult immunodeficient mice, while 100,000 CD133-negative cells failed [30]. Since then, numerous studies have related the expression of CD133 with stem-like properties in pHGG [61,62] and in DMG [63,64,65]. However, contrasting results obtained with adult GBM samples have shown CD133-positive and negative cells as able to convert into each other [66,67] and equally endowed with tumor initiation capacity [66,68], which dramatically requestioned the relevance of CD133 as an accurate marker of adult GSCs. Whether CD133 more soundly correlates with the GSC phenotype in pHGG compared to adult HGG remains to be addressed.

#### 3.1.2. Bmi-1

CD133 is usually co-expressed with other proteins, including the B cell-specific Moloney murine leukemia virus integration site 1 (Bmi-1). Bmi-1 is a member of the Polycomb repressor complex (PRC) 1, which mediates gene silencing by regulating the chromatin structure, and which is required for the self-renewal of both normal and cancer stem cells [69].

High expression levels of Bmi-1 have been reported in pHGG [45,62] and in DMG [63,64,70]. The silencing of Bmi1 expression in a pHGG patient-derived orthotopic xenograft (PDOX) model decreased cell proliferation in vitro and inhibited tumor formation of both CD133-positive and CD133-negative subpopulations *in vivo.* However, gene expression profiling revealed a downregulation of different molecular targets of Bmi1 in CD133-positive compared to CD133-negative cells, which consisted of a novel set of core genes whose modulation impaired tumor initiation [62]. The Bmi1 inhibitor PTC-209 [71] reduced DMG cell proliferation, auto-renewal, migration, cell cycle, and telomerase activity. PTC-209 treatment affected the Rb pathway (which initiates DNA replication during the cell cycle) and increased tumor cell sensitivity to DNA damages caused by radiomimetic drugs [64].

#### 3.1.3. ALDH

Over the past decade, high expression of aldehyde dehydrogenase (ALDH) has been used as a marker for normal stem cells and CSCs in many types of tissues. ALDH is a superfamily of enzymes that detoxify a variety of endogenous and exogenous aldehydes and are involved in resistance to drugs and radiation. In addition, ALDH is required for the biosynthesis of retinoic acid, which participates in both self-renewal and cell differentiation in various stem cells [72].

A recent study identified heterogeneous ALDH expression in different patient-derived DMG H3 K27-altered cell lines [65]. ALDH-positive cells were highly proliferative, demonstrated a capacity to form neurospheres and led to a decreased survival in mice upon orthotopic xenograft, in contrast to ALDH-negative cells. A transcriptomic characterization revealed high mRNA levels of *MYC*, *E2F*, DNA damage repair (DDR), glycolytic metabolism, and mTOR signaling genes in ALDH-positive compared to negative cell lines, which supports a stem-like phenotype. The targeting of MAPK/PI3K/mTOR recapitulated the downregulation of *MYC, E2F*, and DDR genes, diminished glycolytic metabolism in vitro, and inhibited tumor growth in vivo, likely by reducing cancer stemness [65].

#### 3.1.4. L1CAM

The L1 Cell Adhesion Molecule (L1CAM) regulates neural cell growth, survival, migration, axonal outgrowth, and neurite extension during CNS development [73].

The targeting of L1CAM in vitro using shRNA interference in CD133-positive glioma cells (including pHGG) strongly disrupted neurosphere formation, induced apoptosis, and inhibited cell growth. *In vivo*, silencing of L1CAM expression suppressed tumor growth and increased the survival of tumor-bearing animals [74].

#### 3.1.5. Mushashi-1

Musashi-1 (Msi1) is an RNA-binding protein. It is highly expressed in the normal CNS, where it is an important marker of NSCs or progenitor cells [75]. Msi1 modulates Notch signaling and has multiple functions, including maintenance of the NSC state and self-renewal ability, differentiation, and tumorigenesis [75]. Interestingly, Msi1 expression was described in pHGG samples [45,76,77], where it was shown to enhance resistance to chemotherapy [77]. *In vitro* study of pHGG demonstrated that Msi1 promoted the expression of CD44 (see below), therefore co-expressed with MSI1 within recurrence-promoting cells at the migrating front of primary GBM samples. Mechanistically, Msi1 impaired CD44 downregulation in a 3′UTR- and miRNA-dependent manner, by controlling mRNA turnover [76].

#### 3.1.6. Nestin

The neuroepithelial stem cell protein Nestin (NEural STem proteIN) was initially described in NSCs of the developing and adult brain, but is now known to be expressed in a variety of normal and malignant tissues. Physiologically, Nestin is a class VI intermediate filament component of the cytoskeleton, required for cell survival, self-renewal, and mitogen-stimulated proliferation of neural progenitor cells [78]. Often co-expressed with other stem cell markers, such as CD133 and/or Sox2 (see hereunder), high Nestin expression was found in pHGG [30,45,62] and DMG [63,64,70,79,80].

#### 3.1.7. Sox2

In recent years, the aberrant expression of sex-determining region Y (SRY)-box 2 (Sox2) has been detected in a wide diversity of cancers. Sox2 is considered as one of the key founding members of core pluripotency-associated transcription factors. Indeed, it plays a main role in the differentiation of pluripotent stem cells to neural progenitors and in sustaining the properties of neural progenitor cells [81]. As described above, Sox2 is generally co-expressed with other stem cell markers such as CD133 and Nestin in pHGG [45] and DMG [63,70,79,80,82].

#### 3.1.8. Olig2

The oligodendrocyte transcription factor 2 (Olig2) is present, to various extents, in all grades of pediatric and adult diffuse gliomas [83]. Olig2 expression is restricted to CNS, where it influences the proliferation of glial progenitors, the specification of oligodendrocyte progenitor cells (OPC) from neural progenitors or their primitive progenitors (pri-OPC) and the fate switch of OPC-astrocyte by inhibiting astrocytic differentiation in the developing brain [84]. A high expression of Olig2 was co-expressed with other stem cell proteins such as CD133, Nestin, and Sox2 in DMG [63,64,70,79,80,82].

#### 3.1.9. Nanog

Nanog is a transcription factor that includes a DNA-binding domain, and is part of the core regulatory network that suppresses differentiation and maintains pluripotency [85]. Furthermore, Nanog expression closely correlates with stem-like traits in some malignant conditions, including adult GBM [86]. Nanog expression was evaluated in 24 post-mortem DMG tumors, together with other putative GSC protein expression. Whereas Sox2 and Olig2 were expressed in almost all samples, all cases were negative for Nanog [70]

#### 3.1.10. CD44

CD44 is a membrane glycoprotein, known as a receptor for hyaluronic acid and is involved in diverse cellular processes including cell motility, proliferation, apoptosis, and angiogenesis. CD44 has been associated with a stem-like phenotype in adult GBM [87], although disrupting CD44 has later been proposed to increase stem-like phenotype [88,89]. Many studies have investigated its intricate role(s) in tumor cell invasion, proliferation, and resistance to chemoradiation therapy [90].

In pHGG, postmortem analysis of DMG K27M-altered tumors with supratentorial dissemination revealed an upregulation of CD44 correlated with c-SRC activation in multiple foci, which most likely contributed to invasiveness [91]. CD44 also has been shown highly expressed in patient-derived DMG cell cultures [92,93].

#### 3.1.11. CD15

CD15 is a trisaccharide 3-fucosyl-N-acetyllactosamine, also known as stage-specific embryonic antigen 1 (SSEA1). It has been confirmed to be prominently upregulated in normal cells such as neutrophils, macrophages but also NSCs, as well as in several cancers including adult GBM [94]. Studies have revealed high expression of CD15 in patient-derived DMG H3 K27-altered cells [92] and in pHGG [95] models, without deeply investigating its functional role(s).

### 3.2. Recapitulation of Patient Tumor Features

Among a large majority of non-stem tumor cells, GSCs have long been described as essential for forming a tumor that is molecularly and phenotypically close to the original tissue (Figure 2A). It was demonstrated for the first time in 2004, when the xenograft of CD133-positive pHGG recapitulated the histopathological features of the patient’s initial tumor after serial transplantation [30].

*In vitro*, the establishment of patient-derived GSC cultures in a serum-free medium promotes enrichment in stem-like cells, and has long been demonstrated as the most reliable culture procedure for retaining initial patient tumor features [96,97]. In that sense, GSC cultures from pHGG and DMG H3 K27 altered tumors were shown to be proliferative, positive for several stem cell markers, able to differentiate, and endowed with a tumorigenic potential. Importantly, these cell lines accurately reflected the tumor patient in terms of methylation pattern, copy number alterations, and DNA mutations [98].

A paper showed that patient-derived orthotopic xenograft models of pHGG closely resembled their respective GSC line at the molecular level, and recapitulated patient tumor methylation profile and clinical outcome [99]. Another recent paper reports the establishment of 21 PDOX models as well as 8 matched cell lines from various pHGG groups (including DMG H3 K27-altered, diffuse hemispheric glioma H3 G34-mutant and diffuse pHGG, H3-wildtype and IDH-wildtype). The histology, DNA methylation signature, DNA mutations, and gene expression pattern of the patient tumors from which these models were derived were replicated in great majority [100]. However, it has to be reminded that neither GSC cultures nor PDOX models allow to precisely address whether GSCs in situ are the only cells that are responsible for tumor recapitulation.

### 3.3. Models for GSC Generation and Maintenance in pHGG

In adult GBM, the generation of tumors has first been suggested to follow a hierarchical model where a GSC population is at the source of mainly unidirectional state changes towards a more differentiated progeny with a more restrictive profile [101]. Recently, genetic barcoding of freshly isolated adult GBM cells transplanted into PDOX mouse models further provided evidence for a conserved proliferative hierarchy, in which slow-cycling tumor stem-like cells give birth to a quick cycling, self-renewing, progenitor-like population [102].

Such a hierarchical model has been challenged in the last decade, where a model for stochastic evolution of tumors has progressively been introduced. Recent studies have shed light on the strong plasticity of adult GBM cells that dynamically transit from one cellular state to another [56,103], in response to microenvironmental cues, cell–cell interactions, or therapeutic pressure. Enriched expression of several GSC markers (e.g., CD44, CD133, Nestin) has been associated with distinct cellular states, confirming their specificity to a transitory cell status rather than to a cell entity [104,105].

Importantly, this extensive plasticity of adult GBM cells appears more restricted in adult low-grade (IDH mutant) gliomas [103] and DMG H3 K27-altered [56]. Indeed, scRNA-seq analysis of primary DMG H3 K27-altered described that tumor cells are mostly constituted of cells that resemble oligodendrocyte precursor cells (OPC-like), which display an enhanced proliferative and tumor-propagating capacity compared to the minority of more differentiated cells (astrocyte-like and oligodendrocyte-like) [56] (Figure 2B). This H3K27M hierarchy suggests a more narrowed tumor evolution compared to adult GBM (reviewed in Suva et Tirosh [104]), and also provides clues about the putative cell of origin in these tumors (see below). Consistently, a longitudinal study of pHGG genomic profiles revealed a proliferative hierarchy of tumor cells, with slow-cycling cells giving rise to more “proliferating” then “differentiated” cells [106].

## 4. The Subventricular Zone as a Key Actor in pHGG:

In the adult brain, two well-described neurogenic zones host both NSCs and GSCs: the SVZ, located in the walls of the lateral ventricles, and the subgranular zone (SGZ) of the hippocampal dentate gyrus. A limited but sustained neurogenesis is ongoing within those two areas, even in adults [107]. Furthermore, the SVZ has been shown to offer to GSCs a particular microenvironment participating in their resistance to chemo- and RT [108,109]. These observations have raised the question of the cell of origin of HGG, on the one hand, and of the role of GSCs in HGG recurrence, on the other hand. These questions are particularly important in children, in whom the absolute numbers of proliferating cells vastly outnumber those in the adult brain (for a review, see Baker et al. [110]).

Here, we will present data dealing with the cell of origin in pHGG. We will then discuss the potential role of the SVZ in pHGG recurrence. Finally, we will summarize data evaluating how it could impact the survival of patients with pHGG.

### 4.1. The Cell of Origin in pHGG

Identifying the cell-of-origin responsible for glioma development has been a challenge for several years. In that attempt, Parada and collaborators have modeled GBM development in genetically-engineered murine models based on the deletion of p53, Nf1, and Pten tumor suppressors in restricted cell populations (under lineage-specific transcriptional control). Such studies allowed to point out NSCs located in the SVZ as usual suspects for HGG initiation [111], which was recently based on the genetic analysis of tumor material with matched SVZ samples that carry low-driver mutations [112]. Further investigations introduced the concept of a non-unique origin, and showed that lineage-specific gliomas could respectively arise from mutations in the NSCs from the SVZ or OPCs. Both types of gliomas have a distinct transcriptomic profile, methylation profile and functional properties [113], and were shown as differentially sensitive to chemotherapeutic drugs, supporting the consideration of different treatment opportunities based on the tumor profile. With regard to pHGG, a subset of Nestin/SOX2-positive cells also expressing the OPC-related protein Olig2 have been proposed as the tumor-forming population in DMG H3 K27-altered. These cells were detected in the pons of children, and their abundance correlated with the location and timeframe of glioma development [114]. Such finding is corroborated with the recent single-cell transcriptomic data provided by Filbin et al., suggesting DMG H3 K27-altered as enriched in OPC-like cells with great tumor-propagating potential [56]. On the other hand, diffuse hemispheric glioma, H3 G34-mutant were recently demonstrated as developing from GSX2+ interneuron progenitor-like cells (IPC-like) from the SVZ [115]. Altogether, it seems that diverse cell types may give rise to tumors with high genomic and phenotypic diversity, which stresses the importance of developmental programs in determining glioma subtype (for reviews, see Baker et al. [110], Alcantara Llaguno & Parada [111]).

### 4.2. The Potential Role of GSCs and SVZ in pHGG Recurrence

In adult GBM, both experimental and clinical data suggest that GSCs could migrate from the tumor mass towards the SVZ, where they could escape therapies and be involved in GBM recurrences (for a review, see Lombard et al. [116]).

In children, the cellular and molecular mechanisms mediating a possible migration of glioma cells towards the SVZ are largely unexplored. In 2014, Caretti et al. evaluated the SVZ spread of DMG, based on an autopsy series of 16 patients. They found that, in 10 of the 16 patients, there was contact or invasion of the SVZ during the disease, raising the question of the tropism of DMG cells for the SVZ neurogenic niche [117,118]. More recently, Qin and collaborators established two cultures of a pontine DMG at the time of early postmortem autopsy: one from the tumor in the pons and one from the tumor in the SVZ. Using an orthotopic mice model, they showed that SVZ-derived DMG cells injected in the pons migrated towards the SVZ in response to chemoattractant signals secreted by SVZ-hosted neural precursor cells. They identified pleiotrophin (PTN), and its three binding partners—secreted protein acidic and rich in cysteine (SPARC), SPARC-like protein 1 (SPARCL1) and heat shock protein 90B (HSP90B)—as key mediators of this chemoattractant effect [79] (Figure 2C).

### 4.3. The Prognostic Impact of an SVZ Involvement in pHGG

As described above, the specific environment offered by the SVZ to the NSC and GSC could be involved in both the genesis and the recurrence of the pHGG. The clinical significance of a contact between the tumor and the SVZ at diagnosis or at recurrence has been widely studied in adults (for a review, see Lombard et al. [116]).

To the best of our knowledge, only one study evaluated the impact of a SVZ involvement by the tumor at diagnosis on the survival of children with pHGG [118]. In this study, SVZ involvement (SVZ+) was defined by a contact of the contrast-enhancing part of the tumor with the lateral wall of the lateral ventricle on preoperative post-gadolinium T1-weighted magnetic resonance imaging. A total of 63 children and adolescents with pHGG (excluding DIPG) were included (29 SVZ- and 34 SVZ+). Clinical features were similar in both study groups except for more midline location and a higher tumor volume in SZV+ tumors. In the univariable analysis, near- or gross-total resection and seizure presentation were correlated with increased overall survival, while SVZ+ tumors were associated with decreased overall survival (508 days in SVZ+ vs. 981 in SVZ −, HR = 1.94, 95% CI 1.03–3.64, *p* = 0.04). In a multivariate analysis considering the tumor volume and the degree of resection, SVZ+ tumors remained significantly associated with decreased survival (HR = 1.94, 95% CI 1.03–3.64, *p* = 0.04) [118]. These results suggest that tumor contact with the SVZ is a general negative prognosis marker in non-DIPG pHGG and invites biological investigations to better consider, study, and understand the role of SVZ in glioma pathobiology.

## 5. From the Concept of Pediatric GSC to Targeted Therapies

Standard treatments, such as surgery and RT, have been used to specifically target the SVZ in adult GBM and several studies have evaluated their impact on survival (for a review, see Lombard et al. [116]). In pHGG, it is unknown whether irradiating the SVZ in complement to the tumor mass could improve the survival. However, when administered at a young age, RT could lead to possible neurocognitive deficits related to NSC alterations [119,120]. More recently, innovative strategies specifically targeting the GSCs have been developed in adult HGG [121]. In children as well, different methods of GSC targeting have been evaluated. Here we review the various opportunities for targeting GSCs in pHGG via the targeted inhibition of specific signaling pathways (5.1) and the use of oncolytic viruses (5.2).

### 5.1. Targeting of the GSC Signaling Pathways and Metabolism

Several signaling pathways are shared between NSCs and GSCs (Figure 2D) and could play the main role in GSC maintenance.

The Hedgehog (Hh) signaling pathway is highly conserved during evolution and is a key regulator of embryonic development processes, including cell differentiation, proliferation, and tissue patterning [122]. Using an in vitro model derived from early postmortem DMG tissues, Monje et al. showed that the Hh signaling was active in DMG cells and could be involved in the transformation of a potential cell of origin located in the ventral pons [115]. Exposure of DMG cells to the Hh pathway antagonist KAAD-cyclopamine significantly reduced their ability to generate neurospheres [114].

The NOTCH signaling is involved in NSC proliferation, survival, self-renewal, and differentiation [123]. It plays multiple roles in both CNS development and brain tumor biology [124]. High levels of NOTCH receptors, ligands, and downstream effectors are expressed in DMGs, where its inhibition reduces DMG growth, induces apoptosis, and increases sensitivity to RT [125].

The MAPK and the PI3K-Akt signaling pathways play a role in cell proliferation and differentiation, survival, and gene expression [126]. Both pathways are implicated in many cancers [127,128], including pHGG and DMG [129,130]. Inhibition of MAPK/PI3K/mTOR reduced the stem-like phenotype of ALDH-positive DMG cells [65]. Preclinical single-agent targeting of PI3K/mTOR pathway in DMG has seemed promising [131] and dual inhibition of MAPK and PI3K/mTOR pathway in DMG has shown to induce synergistic antitumor effects [130,132]. Long-term survivors have been reported with personalized therapy targeting the PI3K pathway [133].

Finally, specific therapies can also be oriented against metabolic vulnerabilities. Using single-cell RNA sequencing, it was recently shown that DMG H3 K27-altered contains different cell subtypes (namely OPC-like and astrocyte-like cells) with distinct metabolic profiles that can be selectively targeted [134].

### 5.2. Oncolytic Virotherapy

Oncolytic virotherapy has demonstrated a strong efficiency for the treatment of solid cancers, and the first FDA-approved oncolytic virus has recently been introduced for the treatment of melanoma [135]. OVs can be engineered and designed to selectively enter and replicate into cancer cells, e.g., through the presence of a specific receptor on the tumor surface, leading to cell lysis.

Friedman et al. investigated the capacity of a genetically-modified oncolytic Herpes Simplex Virus (HSV) (G207) to infect and kill glioma cells. G207 contains deletions in the γ34.5 and ICP6/UL39 genes that prevent virus killing of normal brain cells. This virus has previously been proven safe in adult GBM patients, with a modest clinical response [136]. They have shown in vitro that CD133-positive and CD133-negative glioma cell subpopulations were similarly responsive to G207-induced cell death, claiming that putative stem-like cells were not resistant to the virus [137]. Another oncolytic HSV (G47Delta, with deletions in the γ34.5, ICP6/UL39, and additional deletion of the immediate-early alpha47 promoter) has been shown to reduce self-renewal of GSC cultures in vitro, and reduced their tumorigenicity. However again, both CD133-positive and CD133-negative cells were equally infected by the virus, stressing the complexity of CD133 in the stem-cell phenotype [138].

Other viruses have been considered for the treatment of pHGG. In 2016, Josupeit and collaborators evaluated the potential of parvovirus H1 (H-1PV). H1-PV is a non-pathogenic small single-stranded rodent DNA virus able to infect and replicate into human cells. They showed that H-1PV efficiently replicated in pHGG neurospheres and induced cytotoxicity in vitro. Nonetheless, H-1PV was able to target both stem-like and differentiated cell subpopulations in neurosphere cultures [139]. Overall, both studies showed that OVs allow the targeting of all glioma cells, including GSCs.

Delta-24-RGD (DNX-2401) is a replication-competent oncolytic adenovirus genetically modified capable of infecting and killing glioma cells, and stimulating an anti-tumor immune response. To enhance potency, an RGD-motif was engineered into the fiber H-loop, enabling the virus to use αvβ3 or αvβ5 integrins to enter cells. These integrins are typically enriched at the surface of tumor cells, including adult GSCs [140] but also in DMG cells [141,142]. The administration of DNX-2401 in mice was proven safe and resulted in a significant increase in survival in immunodeficient and immunocompetent models of pHGG and DMG [142].

Beyond their oncolytic capacity, OVs may elicit an immune response [143] that raises great interest in the treatment of immunologically “cold” tumors such as pHGGs [144]. Very recently, Friedman et al. investigated whether G207 injection triggered a clinical response in pHGG patients [145]. Friedman et al. showed that G207 injection converted “cold” pHGG to immunologically “hot”, which provides great hopes for patient treatment.

Moreover, using mesenchymal stem cells (MSC) as OV carriers improved OV delivery at the tumor site and offered protection from the clearance of the virus by the immune system [146], as recently demonstrated in DMG H3 K27-altered brainstem xenografts in mice. They showed that OV-loaded MSCs could reach the tumor and release the OV. They also reported that the concomitant administration of RT resulted in improved survival compared to OV-loaded MSCs alone [146].

## 6. Conclusions

Despite numerous innovative treatments, the survival of infants and children with pHGG remains poor and did not improve over the last decades. This can be explained by the rarity of pHGG and the low availability of tumor tissues for research purposes, the limited therapeutic approaches due to high risk of neurologic sequelae at young ages, and the absence of recognition of a specific and well-described pediatric entity.

In this regard, the recent publication of the WHO CNS5 represents a major advance in the management of pHGG and has backed more than ever the importance to consider them as complete distinct entities, isolated from their adult counterparts.

For decades, GSCs were studied in adult GBM, providing novel insight into GSC biology that continuously requires further understanding. The study of GSCs was later translated to pHGG, and warrants specific interest. Although the knowledge about their characterization and function is still incomplete, functional and in situ evidence about GSC presence in pHGG is increasingly brought to light. Proteins were identified in pHGG cells as important players in the retention of a stemness state, proliferation, and resistance to therapy. Several studies also show that pediatric GSCs recapitulate the patient tumor phenotype upon xenotransplantation. Recent application of new cutting-edge technologies in pHGG allowed in-depth GSC characterization in situ, depicting a hierarchical model of pediatric GSCs in their native environment. It remains to be seen whether future studies will challenge or consolidate this model. The SVZ has been suggested as a hideout for GSCs, thanks to a peculiar environment that promotes their maintenance in a stem-like state. These findings encourage us to consider these hidden cells as possibly responsible for the origin and the recurrence in future treatment strategies, and further reflect on SVZ targeted therapy. Finally, different signaling pathways have been evidenced in pHGG, and specific targeted therapies are under evaluation.

Altogether, the last years have provided an enriched understanding and remodeling of the concept of GSCs in pediatric-type tumors. In parallel, the important molecular differences that characterize pediatric (vs. adult) HGG have led to the development of novel targeted therapeutic approaches for children (e.g., epigenetic modulation or immunotherapy) (reviewed in [17]). However, the establishment of such children-oriented therapies based on GSC targeting and/or modulation is less advanced. How the recent insight on GSCs could finally translate towards the establishment of therapeutic approaches in pHGG remains to be determined.

Future perspectives will only be possible through close collaboration between the basic science and the clinics, on the one hand, and between the pediatric and the adult fields of expertise, on the other hand, while keeping in mind the specificities of pHGG occurring in infancy and childhood.

## Figures and Tables

**Figure 1 cancers-14-02296-f001:**
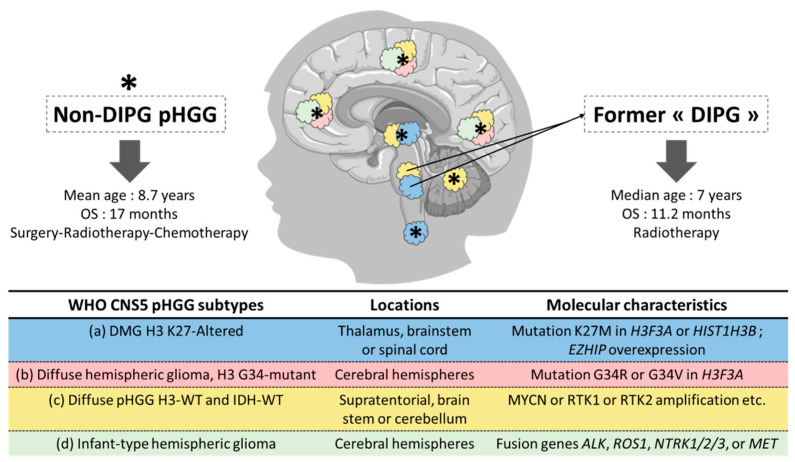
Clinical and molecular characteristics of the four pHGG subtypes according to the WHO CNS5 classification.The recent Fifth edition of the WHO Classification of Tumors of the Central Nervous System has defined, among the dedicated Pediatric-type high-grade gliomas (pHGG) entity, four histomolecular subtypes: (a) the *Diffuse midline glioma H3 K27-altered* (blue), (b) the *Diffuse hemispheric glioma, H3 G34-mutant* (red), (c) the *Diffuse pediatric-type high-grade glioma, H3-wildtype and IDH-wildtype* (yellow) and (d) the *Infant-type hemispheric glioma* (grey). Most of the published data were reported before this recent classification and classically divided the pHGG between diffuse intrinsic pontine glioma (DIPG) and non-DIPG tumors, based on their location. The correspondence between the former, location-based, and the current, histomolecular-based classification is represented on the sagittal brain section using a color code. Abbreviations: ALK: anaplastic lymphoma kinase; DIPG: diffuse intrinsic pontine glioma; DMG: diffuse midline glioma; EZHIP: EZH inhibitory protein; H3: histone H3; *H3F3A*: gene encoding H3.3; *HIST1H3B*: gene encoding H3.1; IDH: isocitrate dehydrogenase; MET: MET proto-oncogene, receptor tyrosine kinase; NA: non-applicable; NTRK1/2/3: neurotrophic receptor tyrosine kinase 1/2/3; OS: overall survival; pHGG: pediatric-type high-grade glioma; ROS1: ROS Proto-Oncogene 1, receptor tyrosine kinase; WHO CNS5: fifth edition of the WHO Classification of Tumors of the Central Nervous System; WT: wildtype. The schematic art pieces used in this figure were provided by Servier Medical art (https://smart.servier.com), 25 April 2022. Servier Medical Art by Servier is licensed under a Creative Commons Attribution 3.0 Unported License.

**Figure 2 cancers-14-02296-f002:**
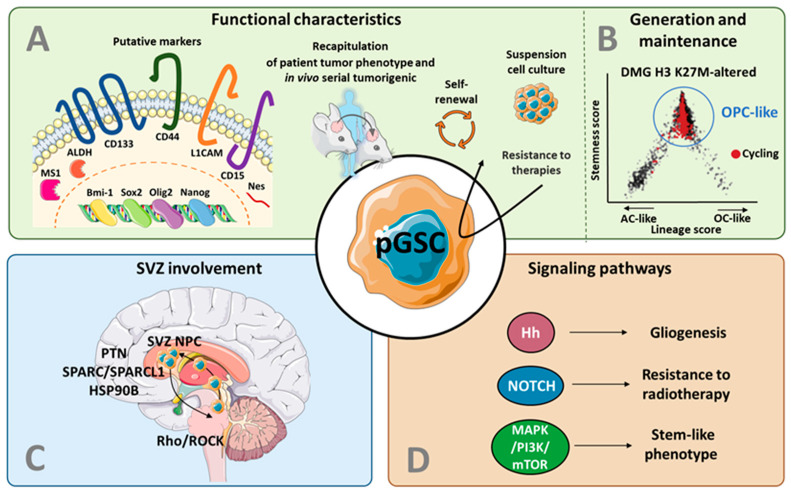
Graphical summary illustrating the functional evidences of GSCs existence in pHGG (**A**), their generation and maintenance in pHGG (**B**), the role of the SVZ (**C**), and the signaling pathways involved in GSC-features (**D**). While there is no clear consensus, glioma stem cells (GSC) are classically defined based on multiple functional criteria, including the expression of different levels and patterns of surface and intracellular markers, the formation of spherical colonies in suspension cultures, a higher resistance to radiation and/or chemotherapy compared to the non-GSCs tumoral cells and the potential to recapitulate the initial tumor heterogeneity (**A**). scRNA-seq analysis of primary Diffuse Midline Glioma (DMG) H3 K27-altered showed that tumor cells are mostly constituted of cells that resemble oligodendrocyte precursor cells, which display an enhanced proliferative and tumor-propagating capacity compared to the minority of more differentiated cells (astrocyte-like and oligodendrocyte-like) (**B**). The subventricular zone (SVZ) is a key actor in pediatric high-grade gliomas (pHGG), as suggested by the recent in vivo demonstration that DMG cells injected in the pons migrate towards the SVZ in response to chemoattractant signals secreted by SVZ-hosted neural precursor cell. This chemoattractant effect depends on the pleiotrophin (PTN) and its three binding partners and is mediated by the PTN receptor protein tyrosine phosphate receptor type Z and the activation of the Rho/Rho kinase pathway (**C**). Several signaling pathways are shared between neural stem cells and pediatric GSC and are involved in gliogenesis, resistance to radiotherapy, and stem-like phenotype of pHGG (**D**). Abbreviations: AC: astrocyte; ALDH: aldehyde dehydrogenase; Bmi-1: B cell-specific moloney murine leukemia virus integration site 1; DMG: diffuse midline glioma; Hh: hedgehog; HSP90B: heat shock protein 90B; L1CAM: L1 cell adhesion molecule; MAPK: mitogen-activated protein kinase; MSI1: mushashi-1; mTOR: mammalian target of rapamycin; Nes: nestin; NPC: neural precursor cell; OC: oligodendrocyte; Olig2: oligodendrocyte transcription factor 2; OPC: oligodendrocyte progenitor cell; pGSC: pediatric glioma stem cell; PI3K: phosphatidylinositol 3′–kinase; PTN: pleiotrophin; ROCK: Rho-associated protein kinase; Sox2: SRY-Box transcription factor 2; SPARC: secreted protein acidic and rich in cysteine; SPARCL1: SPARC-like protein 1; SVZ: subventricular zone. The schematic art pieces used in this figure were provided by Servier Medical art (https://smart.servier.com), 25 April 2022. Servier Medical Art by Servier is licensed under a Creative Commons Attribution 3.0 Unported License. Figure 2B is adapted from Filbin et al. [56].

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
