# Peer review of "Glioma Stem Cells in Pediatric High-Grade Gliomas: From Current Knowledge to Future Perspectives"

_cancers, 2022, doi:10.3390/cancers14092296_

Round 1
Reviewer 1 Report
Review of Da-Veiga et al.: Glioma stem cells in pediatric high-grade gliomas: from current knowledge to future perspectives.
In this review article, the authors provide an update on the underappreciated subject of GSCs in pediatric high-grade gliomas. Although there is some redundancy with the authors’ previous review from last year (Lombard et al., The Subventricular Zone, a Hideout for Adult and Pediatric High-Grade Glioma Stem Cells. Frontiers in Oncology (2021) Jan 26;10614930), there are novel and interesting aspects in this review. This article introduces the new WHO classification for pediatric HGGs, reminds us that functional assays (i.e. serial transplantation assays) but not stem cell markers are essential to determine stem cell properties and attempts to provide an overview on similarities and differences between pediatric and adult HGGs and their respective GSCs. While these insights are interesting for a broad readership, some parts of this manuscript exhibit oversimplifications, suffer from a lack of clarity, and display poor English.
Specific comments:
Large parts of the manuscript suffer from substandard English (subsections 2 and 3.1 to 3.3 exhibit better English with only minor defaults). Please break up long sentences and be clearer about the information/conclusion that is conveyed. As an example, page 3, lines 110-113: “As, in adults, glioma stem cells are more and more recognized as initiating HGGs and being at the onset of relapses after the classical multimodal therapy applied at that age this review will discuss this biological feature in pHGG, remembering that the human brain at that young age is still in development until the end of adolescence.” I strongly recommend having a native English speaker edit these parts of manuscript.
The authors talk about GSCs as tumor initiation cells, but they do not talk about the cell of origin. Is there a hypothesis about the cell of origin in pediatric HGGs? And if so, is it similar or different from the hypothesis in adult HGGs?
The authors rightfully note that pediatric and adult HGGs should be seen as different entities, but then they continue comparing these entities throughout the manuscript. There is substantial knowledge about cancer stem cells in other pediatric brain tumors such as medulloblastoma. I encourage the authors to get inspired by the numerous studies on cancer stem cells in other pediatric brain cancers to discuss childhood-specific aberrant activation of developmental pathways in pediatric brain cancer stem cells.
Subsection 1 is entitled: “From adult to pediatric-type high-grade gliomas”, but this does not fit this section. This section is not elaborating on the differences of adult vs. pediatric gliomas, but rather an informative introduction to the new WHO classification. The figure 1 within this section is also lacking any reference to adult gliomas. Please modify this subtitle according to the topic in this section.
The authors introduce the new WHO classification, but they do not adopt the new classification and instead use the old classification (DIPG vs. non-DIPG) for “the clarity of the present review”. I do understand that most data were generated based on the old classification, but this is exactly what review articles are for: to help a broad readership getting familiar with this new classification. I encourage the authors to adopt the new classification wherever possible.
The bullets/numbering in the first subtitle (1. From adult to pediatric-type high-grade gliomas) are (2.1), (2.2) and so on, but they should be (1.1), (1.2) and so on.
Figures are informative but legends lack a description of the depicted schematics. A single sentence followed by a list of abbreviations is not sufficient for the reader to fully understand the figure. Please elaborate.
Figure 1: the arrow going from the tumor to “former DIPG” is emanating from the yellow tumor, but it should emanate from the blue tumor, because the blue tumor is “DMG H3K27-altered”, or at least there should be 2 arrows emanating from the blue and the yellow tumor.
Figure 2 cannot be understood without a thorough figure legend. For instance, what is the meaning/regulation of the bent arrow indicating “resistance to therapy”? The authors need to provide a detailed figure caption for both figures.
About the selection of potential stem cell markers: It is not clear to me why the authors include CD133, Bmi-1, ALDH, Nestin, etc. but not Nanog, CD44, etc. Are the stem cell markers selected for subsection 3.1 of special interest in pediatric HGGs as compared to adult HGGs?
Paragraph about CD133 (lines 201-216): To my knowledge, Singh et al. 2004 showed that 100,000 cells of CD133-negative cells did not form tumors (and not 10,000 cells as stated by the authors here).
Further within this paragraph, the authors conclude: “Since numerous studies have confirmed the expression of CD133 in non-DIPG pHGGs and in DIPG”. This is again substandard English, but more importantly, this paragraph is an oversimplification that leaves the reader with the impression that CD133 positivity is essential for tumor initiation. They need to mention that in some GBM samples CD133-negative cells are also able to initiate tumors (Beier et al., Cancer Res 2007; Chen et al., Cancer Cell 2010) and that CD133-positive and CD133-negative GSCs were shown to be able to convert into each other within one GBM (Chen et al., Cancer Cell 2010; Wang et al., Int J Cancer 2008). If the authors have evidence that – in deviation to adult GBM – in pediatric HGGs CD133 positivity is essential for tumor initiation, then this would be an important finding and should be further discussed.
Line 313f: “The hierarchical model is abandoned benefit to the stochastic model”. This sentence is highly faulty, in both context and semantic! While there is a vivid discussion about various GSC models recognizing the extremely high plasticity and the cell of origin, the hierarchical model has not been abandoned in its entirety. This is again an example for oversimplification that is misleading the non-expert reader. Also, the following conclusion: “…that pediatric GSC may rather stick to the hierarchical model” is not sufficiently substantiated. Unfortunately, there are more of these oversimplifications that need to be corrected.
Section 4 exhibits very poor English and has some overlap with this group’s recent review in Frontiers in Oncology. Some sentences are very difficult to understand and scientifically incorrect. For instance, lines 378f: “Multiple studies and meta-analyses have also shown that the SVZ (but not SGZ) involvement at diagnosis or at recurrence was linked with poorer OS and could be considered as an independent factor of survival”. The authors need to work extensively on this section before it can be considered acceptable. Also, refrain from using ambiguous/vague terms such as “involvement”. The authors need to be more precise.
Section 5: this is an important section, but it falls short as it mostly represents a list of studies using irradiation and virus-based targeted approaches. This section remains vague and needs a conclusion.
Section 6: this is also an important section that suffers from poor English. The authors point out (lines 463f): “… the importance to consider pHGG as complete distinct entities, isolated from their adult counterparts.” But they do not elaborate on this. In this section (and the previous section) the authors have the opportunity to explain the differences between pediatric and adult GSCs and implications for targeted therapies specifically tailored for children. Unfortunately, section 6 lacks clarity in this regard.
Author Response
Dear Reviewer,
Please find attached the response to the comments made on the manuscript "Glioma stem cells in pediatric high-grade gliomas: from current knowledge to future perspectives". We hope that the modifications and implementations we made will meet your expectations, and thank you for your consideration.
C. Piette and co-authors.

Reviewer 2 Report
I congratulate the authors on a well written paper on the importance/implications of Glioma Stem Cells in Pediatric Malignant gliomas.
I particularly enjoyed the correlation with Glioma Stem Cells in adults and the discussion of the sub ventricular (subependymal) and sub granular zones.
Author Response

(The authors gave the same response as above.)

Round 2
Reviewer 1 Report
The authors have substantially improved the manuscript. They included a paragraph about the cell of origin, improved figure legends, acknowledged the new WHO classification where appropriate, clarified the role of CD133, included additional stem cell markers, discussed more appropriately the hierarchical model and substantially improved sections 5 and 6. Additionally, the English has been improved.